# Bayesian Hierarchical Framework from Expert Elicitation in the South African Coal Mining Industry for Compliance Testing

**DOI:** 10.3390/ijerph20032496

**Published:** 2023-01-31

**Authors:** Felix Made, Ngianga-Bakwin Kandala, Derk Brouwer

**Affiliations:** 1School of Public Health, Faculty of Health Sciences, University of the Witwatersrand, Johannesburg 2193, South Africa; 2Global Biostatistics and Programming, Pharmaceutical Product Development, Part of Thermofisher Scientific, Woodmead, Johannesburg 2191, South Africa; 3Department of Epidemiology and Biostatistics, Schulich School of Medicine and Dentistry, Western University, London, ON N6G 2M1, Canada; 4Département de la Santé Communautaire, Institut Supérieur des Techniques Médicales de Kinshasa, Kinshasa XI, Mont Ngafula, Kinshasa B.P. 774, Democratic Republic of the Congo

**Keywords:** expert judgments, expert elicitation, exposure control categories, the 95th percentile, historical data

## Abstract

Occupational exposure assessment is important in preventing occupational coal worker’s diseases. Methods have been proposed to assess compliance with exposure limits which aim to protect workers from developing diseases. A Bayesian framework with informative prior distribution obtained from historical or expert judgements has been highly recommended for compliance testing. The compliance testing is assessed against the occupational exposure limits (OEL) and categorization of the exposure, ranging from very highly controlled to very poorly controlled exposure groups. This study used a Bayesian framework from historical and expert elicitation data to compare the posterior probabilities of the 95th percentile (P95) of the coal dust exposures to improve compliance assessment and decision-making. A total of 10 job titles were included in this study. Bayesian framework with Markov chain Monte Carlo (MCMC) simulation was used to draw a full posterior probability of finding a job title to an exposure category. A modified IDEA (“Investigate”, “Discuss”, “Estimate”, and “Aggregate”) technique was used to conduct expert elicitation. The experts were asked to give their subjective probabilities of finding coal dust exposure of a job title in each of the exposure categories. Sensitivity analysis was done for parameter space to check for misclassification of exposures. There were more than 98% probabilities of the P95 exposure being found in the poorly controlled exposure group when using expert judgments. Historical data and non-informative prior tend to show a lower probability of finding the P95 in higher exposure categories in some titles unlike expert judgments. Expert judgements tend to show some similarity in findings with historical data. We recommend the use of expert judgements in occupational risk assessment as prior information before a decision is made on current exposure when historical data are unavailable or scarce.

## 1. Introduction

Coal is the second largest energy source in the world, contributing a quarter of the world’s energy sources, and the consumption of coal has been on the rise in recent years. By 2040, worldwide coal consumption is expected to rise at a rate of about 1.5% per year [1]. South Africa is the fourth largest coal producer in the world, employing more than 92,000 workers in 2019 [2]. Coal mining areas in South Africa are geographically found in the KwaZulu-Natal, Mpumalanga, and Limpopo provinces. In coal mining activities, coal dust is produced. Inhalation of coal dust is highly associated with the development of Coal Mine Dust Lung Disease (CMDLD). CMDLD is life-threatening to coal mine workers in developing countries [3,4].

A study by Attfield and Seixas [5] defined the risk of developing CMDLD, a result of over-exposure to coal dust during work-life in the coal mining industry which depends on different work activities. Various work activities generate dust in the coal mining industry. Dust also results from poor organization or lack of best decisions in control of exposure to coal dust. Such activities may include inadequate dust disposal techniques, poor transport systems, coal spillage, and dry mine conditions, amongst others. To improve occupational risk assessment and compliance testing, a homogenous exposure group (HEG) was established. The HEG, according to the Code of Practice (CoP) by the South African Department of Minerals and Energy, is constituted by mine, activity area, and incoming air. Within each HEG, different job titles may be found, increasing the variability of exposure to coal dust [6].

The advancement in technology in recent years means that the coal mining industry has adequate control methods for reducing coal dust overexposures among workers. It is important to regularly check compliance with the occupational exposure limit (OEL) and associated exposure control categories (see Appendix A). Bayesian analysis has become popular for evidence-based policy decision-making. For occupational hygiene applications, Bayesian statistics were used to find highly exposed HEG or job titles. The advantage of Bayesian analysis is that earlier or prior data, when available, can be used to update and improve new measurements using Bayesian inference [7]. The prior distribution can be obtained from solid information (informative) from subjective expert data and/or expert elicitation, or from a non-informative source which does not influence the current data.

Expert elicitation is a systematic approach that combines subjective views from experts on exposure where there is uncertainty because of a lack of or insufficient data [8]. An expert’s experience and knowledge in a particular subject area are the key factors in the elicitation process. Informed decision-making can be made with higher confidence in industries when experts who routinely see and analyze changes in environmental conditions participate in risk assessment. For instance, the United States Environmental Protection Agency uses an elicitation process to solve uncertain problems [9]. Walker et al. (2001) showed that subjective knowledge about the unknown subject can be elicited from the expert for environmental exposure assessment [10]. In occupational hygiene, expert elicitation has been widely used. For example, expert elicitation was used to develop job exposure matrices to categorize job titles into exposure levels [11]. Job exposure matrices can be used when individual exposure data are unavailable or difficult to obtain due to regulatory or privacy measures. Other studies have used elicitation to assess the health effect of fine particles of air pollution [12,13]. Bayesian application of expert elicitation is based on the subjective probabilistic judgement of the previous or present value of an unknown or uncertain quantity [14]. The Bayesian framework can assign exposure probability distributions by incorporating parameters obtained from the expert elicitation, and this is now easy to apply in occupational hygiene [15]. One other benefit of using the Bayesian framework is that it allows for uncertainty in the assessment of compliance to exposure limits, which improves holistic decision-making. Thus, this study aimed to use the Bayesian framework to compare the posterior (after the data were observed) probabilities of the 95th percentile of the exposure distribution (P95) of the non-informative and informative prior from expert elicitation and subjective expert data.

## 2. Methods

### 2.1. Study Design and Data Collection

The study was a survey collection of respirable coal mine dust exposure in underground coal mines in South Africa. The coal mine workers were only males. Workers included in this study were from 10 job titles. The inclusion criteria for the selection of these job titles were based on the availability of previous or historical (past) exposure data and current data. Job titles must have exposure from both the previous and current or the latest year to be included in the analysis. The data from the previous year were used as prior data. Personal sampling of respirable dust was conducted by using cyclones attached to a sampling pump followed by gravimetric analysis [16]. Further details on sampling are available from the earlier paper [6]. Exposures were measured over a full shift and expressed as eight-hour time-weighted average dust concentration (TWA8h concentration).

### 2.2. Expert Elicitation Process

In this study, a modification of the “Investigate”, “Discuss”, “Estimate”, and “Aggregate” (IDEA) structured protocol was used [17]. The first step in the IDEA protocol is the background information on experts, followed by an investigation where experts are requested to individually answer questions and provide reasons for their answers. During the discussion, an expert was shown anonymous answers from each participant to resolve varying interpretations of the questions. Each expert had to estimate for the second and final time. During the post-elicitation process, aggregation of the data takes place, where the mean of experts’ second-round responses is obtained. Many studies have adopted the IDEA protocol seamlessly. The protocol was evaluated in public health, ecology, and conservation studies, and recently in the US Intelligence Advanced Research Projects Activity [18,19]. The advantage of the IDEA protocol is that it minimizes bias resulting from overconfidence, availability, and representativeness. It is also possible to apply for remote elicitation, which is cost-effective. We invited a group of experts to take part in the study. The experts were given time to investigate and understand the questions, and then give their feedback on the questions. The modification from the IDEA protocol is that we did not allow experts to receive feedback on their judgements from other experts. The experts were not encouraged to discuss their results to provide second estimates. We instead adopted and simplified the IDEA protocol four-step elicitation questions in Appendix B. The advantage of the modification is that it prevents more experienced expert(s) from influencing the discussion which may skew the final judgements toward just one dominant expert. The expert elicitation process was done remotely.

In the four-step process we used in this study, the credible intervals (range) were the minimum and maximum probability of exposure in each of the exposure categories. These ranges were standardized using linear extrapolation [18,20].

The minimum standardized percentage: B − ((B − L) × (S/C))

The maximum standardized percentage: B + ((U − B) × (S/C))

Where B is the best possible guess, L is the minimum percentage, U is the maximum percentage, S is the level of credible intervals to be standardized, and C is the level of confidence (how sure is the expert). The distributions were truncated for extreme values if the adjusted credible intervals fall outside the probability bounds of 0 and 100%. The credible intervals were standardized to reduce overconfidence.

The expert’s judgements were combined to form a joint probability distribution. The distribution was used as an informative prior, updating current data from respective job titles to produce posterior estimates when compared to the OEL. The individual expert probability distribution can be mathematically or behaviourally aggregated to produce the joint distribution [21]. For this study, we used equal weighting to aggregate the data.

#### Selection of Experts

Experts were invited through email to take part in this study. The average year of experience was 13 years. The experts were occupational hygienists, mine ventilation engineers, supervisors, mine engineers, and geologists. The experts were found by the lead of occupational health and hygiene in the coal mining industry. The identified experts then recommended other experts through snowballing. All experts who accepted the invitation were asked to voluntarily take part. Experts had a range of experience associated with various responsibilities and/or connections to occupational health and safety in coal mining.

Before the elicitation, experts were trained on the uncertainty, the P95, and an outline of the tasks to be performed. They were also provided with an information sheet to obtain their professional background in the field. Each job title was described in detail to the experts and an example of how the probability of exposure to each exposure category was shown to familiarize them with the elicitation. An expert was allowed to consult at any time if he/she is confused or lost in the elicitation. According to Highlights of the Expert Judgement Policy Symposium and Technical Workshop conference, held in the year 2006 [22], a sample size between six to twelve experts is recommended for expert elicitation. In this study, a sample size of six experts was included.

### 2.3. Statistical Methods

Statistical analysis was conducted in STATA version 14.1 [23], and R version 4.1.1 (R Core Team, Auckland, New Zealand), using RStan and bayestestR packages for the Bayesian model [24,25,26]. A summary of the current and historical data was presented as arithmetic mean (AM), standard deviation (SD), geometric mean (GM), and geometric standard deviations (GSD). Each HEG is said to be unique [6], however the same job title and work activity can be found in different HEGs. The median difference in job title exposure across HEGs was assessed by using the Kruskal–Wallis Test, a non-parametric version of the analysis of variance (ANOVA). A job title with a statistically significant *p*-value of less than 0.05 at the 95% confidence interval was not selected for the Bayesian analysis, as this would mean the exposure distribution is different across HEGs. Such a job title is not regarded as having a similar exposure profile.

#### 2.3.1. Producing Joint Distribution from Experts’ Elicitation

We used the Sheffield Elicitation Framework (SHELF) tool to produce the joint probability distribution. SHELF is an open-source tool developed by the University of Sheffield [27].

The SHELF tool finds the distribution that best fits the expert’s elicited judgement. The SHELF tool works by using lease squares procedures by finding a distribution that best fits the given expert’s data inputs by minimizing the sum of squares of the residuals. Recall that occupational exposure data are best fitted by a lognormal distribution. In this study, we used lognormal distribution to produce our prior means and variance for the Bayesian framework. In the SHELF tool, we used the quartile method to elicit our expert’s judgement in a continuous quantity [28].

The Quartile methods ask for experts’ median and quartiles. The quartiles are presented as lower (Q1) and upper quartiles (Q3). The lower (L) and upper (U) limits during the elicitation were 0 and 100%. The Q1 is a value between L and M, where M is the median value, while Q3 is a value between M and U. In the expert’s elicitation questionnaire in Appendix B and Appendix C, we collected the expert’s best guess in form of percentages to distribute the P95 grouping according to the exposure categories. Additionally, we asked about their minimum and maximum percentages on the probability of the P95 coal dust exposure. This was important to capture the expert’s uncertainty when they may not be sure about their best guess. Therefore, in the Quartile methods, the best guess was used as the median, while their minimum and maximum percentages were represented as the Q1 and Q3, respectively. The cumulative probabilities were set at 0.25, 0.5, and 0.75.

A minimum informative non-parametric distribution then spread the mass uniformly between estimates to make sure the joint distribution fully is the expert assessment [29]. The arithmetic mean of the lognormal mean and variance was obtained with the assumption that each expert contributed equally. The mean and variance were then used prior to the Bayesian framework for the exposure compliance assessment.

#### 2.3.2. Bayesian Framework

Consistent and similar coal dust exposure from expert judgements data of the job titles were used to update current monitoring data. The posterior geometric mean (GM), geometric standard deviations (GSD), P95, and the probabilities of finding the P95 exposure in each of the exposure control categories were obtained. For the prior specification, we randomly selected a prior sample size of five out of the six experts’ data collected, as recommended from earlier studies for occupational exposure assessment [30,31]. From the occupational exposure perspective, the prior sample size should be from 10% to 40% of the current data so that the posterior distribution learned more from the current likelihood data than from the prior distribution. Therefore, the sample size of five was used to keep the focus of the posterior distribution on the current data, as the sample size of the current data increases. In Bayesian statistics, the posterior distribution is a compromise between the information from the prior and the current data, but the distribution must be seen from the current data to a good measure as the sample size increases [32]. For the likelihood function, we took all the available current monitoring data.

#### 2.3.3. Model Specification Using Current Likelihood Data

The likelihood was specified as µ and σ which is the GM and GSD, respectively. The likelihood function is presented below in Equation (1).
(1)$$\prod_{i=1}^{n}LN\left(y_{i}|\mu ,\sigma^{2}\right)=\prod_{i=1}^{n}\frac{1}{y_{i}\sigma \sqrt{2\pi }}exp\left\{-\frac{1}{2}\frac{(logy_{i}-{\mu )}^{2}}{\sigma^{2}}\right\}$$$y_{i}$ is the log-normal (LN) current monitoring data, n is the number of observations. The OEL exposure categories were added as a random variable in the model directly to produce the posterior probability distribution of the P95 to each of the categories [30,31].

#### 2.3.4. Model Specification Using a Non-Informative Prior Distribution, and Informative Prior from the Historical Data and Expert Judgements

The non-informative prior follows a uniform distribution with GM and GSD.

For the GM, $\mu =LnGM\sim Unif\left(a_{\mu }b_{\sigma \mu }\right)$, and for the GSD, $\sigma^{2}=LnGSD\sim unif\left(a_{\sigma }b_{\sigma }\right)$, where a and b are the lower and upper bounds of the prior distribution, respectively.

For the informative prior, the GM μ takes the form as shown below in Equation (2).
(2)$$\mu ∼Norm({\overset{-}{y}}_{0},s_{y0}^{s}/n_{0})$$
where ${\overset{-}{y}}_{0}$ and $s_{y0}^{s}$ are the prior mean and variance, respectively, and $n_{0}$ is the prior sample size.

For the GSD, the variance is given by Equation (3).
(3)$$\sigma^{2}\sim IG\left(\frac{n_{0}-1}{2},\frac{\left(n_{0}-1\right)s_{y0}^{2}}{2}\right)$$

For $n_{0}>1$, where IG (a, b) is an inverse gamma distribution in Equation (4) with shape parameter a, and scale parameter b

Therefore,
(4)$$a=\frac{n_{0}-1}{2}\;{b=2/(s}_{y0}^{2}\times (n_{0}-1))$$

Further details on the prior specification and full conditional for μ and $\sigma^{2}$ are available at Made et al. [33], and Quick, 2017 [31].

#### 2.3.5. The Sensitivity Analysis for the Parameter Space

In this study, we placed restrictions on the lower bounds of the μ and $\sigma^{2}$ and P95 using the suggestions from Bayesian decision analysis (BDA) [15]. For the upper bound, we took motivation from Quick, 2017 [31], and allowed it to vary iteratively. The upper bound was not placed on any restriction to avoid being unfairly skewed towards a more favorable or unfavorable result. Despite this, we believed that the use of the parameter space might misclassify an exposure to a wrong exposure category. Therefore, we randomly took two job titles, namely shuttle car operator and pump attendant. We changed the parameter space and compared their findings.

In shuttle car operator, the GM and GSD’s lower bound was set at 0.00001, assuming that the coal dust exposure cannot be zero. Their upper bound is set to be just above the GM and GSD values calculated in this study data. The GM is 1.08 so the upper bound was placed at 1.09; the GSD is 2.62 and upper bound was placed at 2.63, to make sure the exposures estimate fall within.

For the pump attendant, we placed the upper bound of the GM to be 0.42, just above the actual value of 0.41, while GSD was placed at 3.42 above the actual value of 3.41.

We also ran analysis on these two job titles without placing any restrictions on the parameter space. In other words, no lower and upper bounds were specified.

The full posterior conditional distribution was drawn using the Markov chain Monte Carlo (MCMC) of the Gibbs sampler [34]. The Gibbs sampler was used because of computational flexibility. The Gelman–Rubin convergence diagnostics was used to assess the stationary of the posterior distributions including the reliability of the posterior samples. Please refer to Made et al. [33] for more details on the Gibbs sampler and convergence diagnostics.

## 3. Results

A summary of the job titles and their corresponding exposures are presented in Table 1. A total of 10 job titles were included in this study. The job titles were chosen on the basis that they have both monitoring (current) data and earlier (historical) data in the HEG they belong to. There are 455 observations in the monitoring data and 108 from historical data. Six job titles in the current data showed high exposure variability according to the GSD (GSD greater than 3), while from the historical data, eight job titles showed high exposure variability according to the GSD. All the job titles showed statistically non-significant differences in median exposure variation across their parent HEGs (*p* < 0.05).

Table 2 shows the posterior GM, GSD, and P95 from the Bayesian analysis. The GM from the non-informative and informative prior distribution were all below the OEL. Only one job title had a GM above the OEL in the Bayesian findings generated by informative prior from expert judgments. For variability of exposures, four and five job titles showed high variability according to the GSD in the non-informative and informative prior from historical data, respectively. The posterior GSD from the expert judgments showed that six job titles had a high variability of exposure. Three of the job titles showed consistently high variability of exposures across all the methods.

Figure 1 shows the comparison of the posterior P95 by job titles. The findings from the use of expert judgments as prior indicated higher P95 in all job titles exposure. Only for pump attendants, roofbolt operators, safety officers, and shuttle car drivers, the P95 is lower from historical data than from the non-informative prior distribution. Table 3 estimates the probabilities of grouping exposure in each exposure category. Overall, there were high probabilities (above 75%) of finding exposures in the highest category (4) (above OEL/very poorly controlled). The pump attendants’ exposures had a higher probability of being in exposure category 4 from non-informative prior (79.69%) than historical data (73.09%). However, there were higher probabilities of being in exposure category 3 when using historical data compared to the non-informative. The findings from the experts showed more than a 99.9% probability of all the exposures in the poorly controlled group (exposure category 4).

Table 4 shows the results of the sensitivity analysis. The findings of the parameter space we employed versus putting no restriction on the bound and using different parameter values were compared (see Section 2). Using no parameter space revealed 100% probability of finding the P95 in the highest exposure category. Using parameter values described in the Section 2 showed similar findings with the values inspired by the BDA on the probabilities of locating the P95 in each exposure category. The posterior probabilities when using experts’ judgment as prior were similar (99.94% versus 99.91% for exposure category 4). The findings from the non-informative and informative prior from historical data showed slightly higher probabilities of the exposure in category 3 when using different parameter values than values from the BDA. Since we used a lower parameter space for the lower and upper bound, this resulted in lower probabilities of finding exposures in category 4.

## 4. Discussion

This study aimed to compare the posterior probabilities of grouping the P95 according to the COP exposure control categories or groups. We derived informative prior distributions from expert judgments and historical data for each job title. Since the same job titles were found in more than one HEG or mining area, we included those that were similar in exposure distribution across HEG. This is important, as each HEG could have different exposure control methods that could influence exposure variability. It is only natural that the exposure distribution of a job title should be similar regardless of the area, since they perform the same work activity [6]. This is important, because their findings can be generalized back to the workers that have yet to be selected. In occupational exposure assessment, incorporating expert judgments in a Bayesian framework for exposure categories was first used and proposed by Hewett et al. [15]. The use of expert judgments in this study is necessary because if historical data are unavailable or inadequate, expert knowledge can always be used to answer questions about the quantity of interest. The expert elicitation was conducted using a modified IDEA [17]. We changed the IDEA technique by not allowing expert judgments to be shared among each other. It was good to avoid a more experienced expert having too much influence over the final expert prior to distribution. The confidence level in individual expert judgments was adjusted using the 70% upper credible limit from the Committee of European Normalization approaches [35]. The joint probability lognormal distribution for the prior mean and variance from the individual experts were derived using the SHELF package [27]. The lognormal parameters were generated so that the lower point is the 2.5th percentile and the highest value is the 97.5th percentile. SHELF was used to conduct the elicitation of uncertain quantities in the form of probabilities from a group of experts. Instead of running expert group interaction and sharing information to get a consensus, we only used SHELF to produce the lognormal distribution of the mean and variance of the uncertain quantities. These parameters were used as informative priors in a Bayesian framework.

Our study found that the posterior GM from non-informative priors and the prior derived from historical and expert data were all below the OEL (OEL = 2 mg/m^3^), showing no overexposure risk when considering the GM as a parameter to estimate the risk (Table 2). It is important to know that exposure management decisions are based on the P95 of the lognormal exposure distribution [36]. The posterior GSD showed more job titles had very high exposure variability in the expert prior than in the historical and weak priors. The distribution of the median P95 (95% CrI) across all methods is in Figure 1. The P95 estimates the upper bound of the TWA8h exposures and can be achieved for 95% of the workers [37]. Compared to the other two approaches, the P95 was mostly lower in the non-informative prior. The 95% CrIs were more expansive in the non-informative prior and the prior derived from experts’ judgments. That means that a risk decision may not be taken with high confidence, since the level of uncertainty is high. The reason the experts’ data had a wider 95% CrI may also be due to the different beliefs of experts about their uncertainty and level of confidence.

The posterior probabilities of finding the P95 in a specific exposure control category are shown in Table 3. From the informative prior from historical data, there were at least 75% probabilities of finding the exposures in the highly exposed exposure category 4 similarly to the non-informative prior distribution. The findings from using expert judgments showed more than 95% probabilities of exposures in the highest exposure category. This may show the experts either overstated their beliefs or their exposures were higher. The results from the experts are inconsistent with those from previous studies where qualitative exposure assessments showed underprediction of exposure by experts [38,39]. The results generated by the experts and historical data tend to show more consistency. These higher probabilities of finding the P95 in the highest exposure category when using expert judgments might be attributed to overconfidence on the part of some experts, despite reducing their confidence level by 30%. Some of the GSDs indicated low variability of exposure, but the respective probabilities of locating the P95 were found in higher exposure control categories. These contrasting results might show that incorrect priors updated the posterior P95. Therefore, more sampling might be needed to repeat the process and reach a decision. Our results also contrast with the earlier study, which used HEGs; here, we found that the probabilities of the exposures being in the higher category (poorly controlled exposure) were greater than 95% [33]. This clearly explains why grouping by using a job title can give a truer exposure distribution than grouping according to HEGs. Despite having only some of the data for compliance testing, exposure determinants, e.g., specific activities (often associated with job titles), workplace conditions, and production volume, are highly needed in exposure assessment [40,41]. Therefore, job titles are important to consider as a determinant of exposure variation.

The BDA motivated the lower and upper bounds placed on the parameter space in this study (see Table 4) [15]. The use of bounds in the parameter space is essential to avoid the exposure being classified in an unfavorable exposure category, either in the lowest exposure category, the highest exposure category, or infinity. According to the BDA, the GM and GSD cannot be any values, no matter how large or near-impossible they can be like in frequentist statistics. Thus, the minimum and maximum values were set as bounds for GM and GSD. However, in this study, specifying bounds for the parameter space might misclassify exposure towards the favorable or unfavorable exposure control category. We conducted a sensitivity analysis to compare the bounds adopted in this study with if no bounds are specified, or different bounds are used. The findings suggest that using no bounds tends to place exposures in the poorly controlled exposure category compared to using parameter values in the parameter space. The use of another bound shows comparable results to the one we adopted. It is also natural that exposure measurement values in occupational settings cannot be extreme or at infinity, thus the use of bounds. An earlier study revealed that because of the small sample size of the data, the use of no bound could have placed the exposure in the highest exposure category [31].

We draw strength from this study because Bayesian statistics naturally allow earlier information from historical data or expert judgments to be incorporated into the model with the current data for informed decision-making [7]. Even with a small sample size, the results are robust and easy to interpret, even by non-statisticians. One limitation of the study is that, since we had inadequate data from experts, we could not apply weighted aggregation, which would have improved the accuracy of the prior experts. Experts’ judgments can also be very biased, and their responses are dependent on their knowledge of the activities of a specific job title. Human judgments can be complex and biased. Sometimes experts rely on the Rule of Thumb to make decisions or judgments about unknown quantities, including probability assessments. The Rule of Thumb does not synthesize information like the algorithmic process to arrive at a particular judgment [42,43]. Therefore, biases might be common with the use of expert judgments. Expert training is highly encouraged in an expert elicitation process to minimize biases and improve the cognitive interpretation of the information needed. Finally, it is important to note that Bayesian analysis also offers confidence in whether to use experts’ judgment as a prior or not, as it can compare expert prior updates with the non-informative prior.

## 5. Conclusions

Expert judgments are particularly useful when data are scarce or the available data are inadequate. It is well known that experts’ judgments can be very biased, but the use of a robust method and a sensitivity analysis may minimize the bias. Our findings suggest that prior expert judgments can produce similar posterior probabilities of exposure when compared to historical data. This study seeks to advance the prevention of overexposure to coal dust. Therefore, mine ventilation engineers, occupational hygienists, and safety officers are encouraged to incorporate our findings into their routine risk assessments. Future occupational exposure compliance assessments should be based on this Bayesian framework, where historical data or, if unavailable, experts’ judgments should be used to reach an informed decision.

## Figures and Tables

**Figure 1 ijerph-20-02496-f001:**
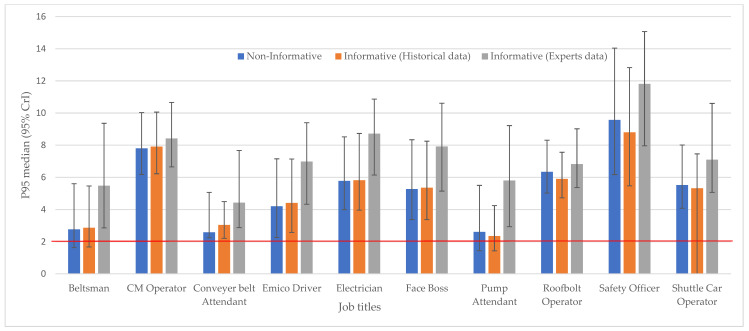
The comparison of the patterns of the posterior median (95% CrIs) of the P95 for non-informative and informative (historical and expert data) distributions of the TWA8h dust concentrations across the job titles. The red horizontal is the SA OEL = 2 mg/m^3^.

**Table 1 ijerph-20-02496-t001:** Summary of coal dust exposure concentration (TWA8h) in mg/m^3^ for the current monitoring data and their corresponding historical past data, and comparison of the median exposure across HEGs.

Data	Year	n	AM	SD	GM	GSD	*p*-Value *
Current data							
Beltsman	2015	18	0.76	0.72	0.48	2.94	0.5694
CM Operator	2018	116	2.08	1.87	1.28	3.30	0.1014
Conveyer belt Attendant	2015	29	0.93	0.75	0.66	2.41	0.0917
Emico Driver	2015	24	1.13	1.02	0.54	5.11	0.3116
Electrician	2015	52	1.55	1.52	0.87	3.75	0.0617
Face Boss	2018	35	1.56	1.76	0.80	3.68	0.8640
Pump Attendant	2016	18	0.74	0.82	0.41	3.41	0.3955
Roofbolt Operator	2018	101	1.77	1.66	1.11	3.03	0.0853
Safety Officer	2018	16	2.78	2.05	2.06	2.45	0.8169
Shuttle Car Operator	2018	46	1.60	1.34	1.08	2.62	0.1615
Historical data							
Beltsman	2009	14	1.07	0.89	0.66	3.31	0.1653
CM Operator	2009	11	2.10	2.05	0.70	8.46	0.1573
Conveyer belt Attendant	2009	7	0.64	0.56	0.42	2.92	0.3012
Emico Driver	2009	6	0.81	0.73	0.43	4.29	0.5319
Electrician	2009	14	1.70	2.03	0.75	5.31	0.3618
Face Boss	2009	22	1.00	0.97	0.46	4.89	0.8495
Pump Attendant	2009	8	0.66	0.69	0.42	2.86	0.3679
Roofbolt Operator	2009	8	1.70	2.58	0.81	3.54	0.5319
Safety Officer	2009	8	0.70	0.56	0.38	4.57	0.4060
Shuttle Car Operator	2016	10	1.3	1.11	0.69	4.59	0.2352

n: sample size; AM: athematic mean; SD: standard deviation; GM: geometric mean; GSD: geometric standard deviation. * *p*-value to test the difference in dust exposure of each job title across HEGs.

**Table 2 ijerph-20-02496-t002:** The median (95% credible interval (CrI)) of the posterior GM, GSD, and the P95 percentiles of the TWA8hdust concentration and the P90 according DMRE CoP (OEL is 2 mg/m^3^).

	Non-Informative			Informative from Historical Data			Informative from Expert Judgments		
Job Titles	GM	GSD	P95	GM	GSD	P95	GM	GSD	P95
	Median (95% CrI)	Median (95% CrI)	Median (95% CrI)	Median (95% CrI)	Median (95% CrI)	Median (95% CrI)	Median (95% CrI)	Median (95% CrI)	Median (95% CrI)
Beltsman	0.48 (0.28, 0.83)	2.90 (2.48, 3.64)	2.77 (1.63, 5.60)	0.47 (0.27, 0.80)	3.01 (2.61, 3.66)	2.87 (1.67, 5.46)	0.84 (0.50, 1.38)	3.10 (2.67, 3.58)	5.48 (2.86, 9.36)
CM Operator	1.28 (1.03, 1.59)	2.99 (2.80, 3.23)	7.80 (6.18, 10.03)	1.25 (1.00, 1.57)	3.06 (2.87, 3.29)	7.91 (6.22, 10.06)	1.37 (1.11, 1.70)	3.00 (2.82, 3.22)	8.42 (6.65, 10.66)
Conveyer belt Attendant	0.66 (0.48, 0.94)	2.59 (2.32, 3.03)	2.59 (2.25, 5.06)	0.64 (0.47, 0.87)	2.57 (2.33, 2.95)	3.04 (2.21, 4.50)	0.80 (0.55, 1.21)	2.84 (2.53, 3.30)	4.43 (2.88, 7.67)
Emico Driver	0.51 (0.26, 0.90)	3.58 (3.09, 4.24)	4.21 (2.26, 7.15)	0.56 (0.34, 0.92)	3.47 (3.03, 4.02)	4.41 (2.58, 7.14)	0.91 (0.58, 1.33)	3.41 (3.08, 3.79)	6.98 (4.33, 9.40)
Electrician	0.86 (0.60, 1.22)	3.18 (2.87, 3.57)	5.78 (3.99, 8.52)	0.83 (0.57, 1.18)	3.26 (2.96, 3.64)	5.82 (3.97, 8.43)	1.31 (0.98, 1.69)	3.14 (2.89, 3.39)	8.72 (6.14, 10.87)
Face Boss	0.79 (0.51, 1.20)	3.16 (2.80, 3.64)	5.27 (3.38, 8.34)	0.75 (0.47, 1.16)	3.28 (2.92, 3.74)	5.35 (3.38, 8.25)	1.19 (0.83, 1.64)	3.13 (2.83, 3.45)	7.92 (5.14, 10.61)
Pump Attendant	0.40 (0.22, 0.74)	3.11 (2.63, 3.91)	2.61 (1.44, 5.50)	0.38 (0.24, 0.62)	2.99 (2.60, 3.65)	2.35 (1.43, 4.25)	0.82 (0.48, 1.32)	3.24 (2.80, 3.70)	5.80 (2.94, 9.22)
Roofbolt Operator	1.11 (0.90, 1.39)	2.88 (2.69, 3.12)	6.34 (5.02, 8.31)	1.04 (0.84, 1.29)	2.86 (2.68, 3.10)	5.90 (4.72, 7.56)	1.17 (0.94, 1.47)	2.91 (2.73, 3.16)	6.82 (5.36, 9.02)
Safety Officer	1.98 (1.24, 2.93)	2.59 (2.27, 3.06)	9.57 (6.17, 14.04)	1.57 (0.88, 2.44)	2.84 (2.50,3.36)	8.80 (5.47, 12.83)	2.30 (1.54, 3.18)	2.68 (2.42, 3.00)	11.81 (7.96, 15.07)
Shuttle Car Operator	1.08 (0.81, 1.45)	2.69 (2.46, 3.04)	5.52 (4.08, 8.01)	1.05 (0.80, 1.38)	2.68 (0.80, 1.38)	5.32 (3.98, 7.46)	1.35 (1.03, 1.84)	2.73 (2.50, 3.06)	7.10 (5.06, 10.60)

P95: 95th percentile; CrI: Credible interval.

**Table 3 ijerph-20-02496-t003:** The estimated exposure category probabilities for the posterior 95th percentile of the TWA8h dust concentration of the non-informative and informative Bayesian frameworks (OEL = 2 mg/m^3^).

HEG	Non-Informative					Informative from Historical Data					Informative from Experts’ Data				
	P95	Category 1	Category 2	Category 3	Category 4	P95	Category 1	Category 2	Category 3	Category 4	P95	Category 1	Category 2	Category 3	Category 4
Beltsman	2.77	0	0.25%	12.07%	87.67%	2.87	0	0.18%	9.52%	90.29%	5.48	0	0	0.06%	99.95%
CM Operator	7.80	0	0	0	100%	7.91	0	0	0	100%	8.42	0	0	0	100%
Conveyer belt Attendant	2.59	0	0	0.38%	99.62%	3.04	0	0	0.40%	99.60%	4.43	0	0	0.02%	99.99%
Emico Driver	4.21	0	0.03%	1.04%	98.93%	4.41	0	0	0.19%	99.81%	6.98	0	0	0	100%
Electrician	5.78	0	0	0	100%	5.82	0	0	0	100%	8.72	0	0	0	100%
Face Boss	5.27	0	0	0	100%	5.35	0	0	0	100%	7.92	0	0	0	100%
Pump Attendant	2.61	0	1.16%	19.14%	79.69%	2.35	0	0.91%	25.99%	73.09%	5.80	0	0	0.06%	99.94%
Roofbolt Operator	6.34	0	0	0	100%	5.90	0	0	0	100%	6.82	0	0	0	100%
Safety Officer	9.57	0	0	0	100%	8.80	0	0	0	100%	11.81	0	0	0	100%
Shuttle Car Operator	5.52	0	0	0	100%	5.32	0	0	0	100%	7.10	0	0	0	100%

Category 1: P95 TWA8h concentration < 0.1 OEL (Very highly controlled); Category 2: P95 TWA8h concentration ≥ 0.1 OEL < 0.5 OEL (Highly controlled); Category 3: P95 TWA8h concentration ≥ 0.5 OEL < OEL (Adequately controlled); Category 4: P95 TWA8h concentration ≥ OEL (Poorly controlled).

**Table 4 ijerph-20-02496-t004:** Sensitivity analysis to compare the parameter values used for the lower and upper bound in this study.

Grouping		Using Parameter Values from BDA				Placing No Restrictions on Bounds				Using Different Parameter Values			
		Category 1	Category 2	Category 3	Category 4	Category 1	Category 2	Category 3	Category 4	Category 1	Category 2	Category 3	Category 4
Pump Attendant	Non-informative	0	1.16%	19.14%	79.69%	0	0	0	100%	0	1.24%	20.85	77.91%
	Historical data	0	0.91%	25.99%	73.09%	0	0	0	100%	0	1.62%	29.11%	69.27%
	Exert judgments	0	0	0.06%	99.94%	0	0	0	100%	0	0	0.09%	99.91%
Shuttle Car Operator	Non-informative	0	0	0	100%	0	0	0	100%	0	0	0	100%
	Historical data	0	0	0	100%	0	0	0	100%	0	0	0	100%
	Exert judgments	0	0	0	100%	0	0	0	100%	0	0	0	100%

Category 1: P95 TWA8h concentration < 0.1 OEL (Very highly controlled); Category 2: P95 TWA8h concentration ≥ 0.1 OEL < 0.5 OEL (Highly controlled); Category 3: P95 TWA8h concentration ≥ 0.5 OEL < OEL (Adequately controlled); Category 4: P95 TWA8h concentration ≥ OEL (Poorly controlled).

## Data Availability

The data presented in this study are available on request from the corresponding author.

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
