# Peer review of "Bayesian Hierarchical Framework from Expert Elicitation in the South African Coal Mining Industry for Compliance Testing"

_ijerph, 2023, doi:10.3390/ijerph20032496_

Round 1
Reviewer 1 Report
In this manuscript, a Bayesian framework with prior information from historical or expert judgements is proposed to assess compliance with exposure limits in the south African coal mining industry. Overall, this paper is not well-organized and there are some deficiencies and mistakes, thus minor revision is suggested. Some questions and comments are listed as follows:
1. Is Table 1 on line 68 and line 268 the same in the paper? If not, please add and modify the serial number.
2. Where is the supplementary material in line 130? Is it the same as in line 68?
3. AM, SD, GM, and GSD should be marked when used for the first time in the paper, such as in lines 170-171. Use the abbreviations directly when using them again. Please check the paper for use.
4. At line 136, the explanation of L variable is missing. The explanation of some variables is also missing in Equations (1)-(2). Please add.
5. Equation 3 should be revised to Equation (3). Do the same for Equation 4.
6. In line 147, what is the weight 1:1:1:1:1?
7. For lines 273-274, choose from "GSD greater than 3" or "GSD>3".
8. In discussion, what do UCL and CEN stand for?
9. Please add the threshold value of OEL, so that the reader can intuitively compare with GM and other parameters.
10. Please revise the abstract to highlight the findings.
11. Is there a difference between the words “non-informative prior”, “prior information”, “informative prior distribution” and “posterior probabilities”? If so, please revise the full paper to avoid ambiguity.
12. The data in the discussion section lacks in-depth analysis.
Author Response
Please have the attached response.

Reviewer 2 Report
The paper is well written - there are a few minor English changes that are needed, but fewer than in most articles.
The use of "expert opinions" as a means of providing input for risk assessments is becoming more common. This is used in QRA studies, especially where some of the specific information required is unknown and/or difficult to quantify. This paper uses the same techniques for a personal safety and exposure application.
I have struggled to understand what the data in Table 1 means. Exposure data normally has a unit, perhaps mg/m3, or mg/m3/8 hour shift. All of the data is dimensionless - it is very unclear what the data means or how to use the data.
For example - in Table 2, the Safety Officer has the highest value - which seems incredible given that the Safety Officer should be most focused on minimizing exposure.
Finally - the study demonstrates that expert opinion can be shown to match measured values. OK - so where does this knowledge go? It would seem that the application is to exposure (and exposure reductions) for locations where measurement is not known. Who would do the and take this forwards to reduce exposure to the workers? Please include a vision of how this work can advance the Prevention message to the workers.

Author Response
Please have the attached response.
